# Scientific Knowledge Map Study of Therapeutic Landscapes and Community Open Spaces: Visual Analysis with CiteSpace

**Yan Han** * and **Yuehui Liang**

Department of Spatial Culture Design, Kookmin University, Seoul 02707, Republic of Korea; yhleung@126.com
* Correspondence: hanyan0536@foxmail.com

**Abstract:** The pursuit of a healthy and comfortable living environment is a key developmental objective for human society. Therapeutic landscapes play a significant role in improving environmental conditions within these spaces. However, current research suggests that there is still much to be explored in this field, particularly in communal open spaces. Based on the Web of Science literature database and using the CiteSpace visualization tool, this study launched a literature review search on the existing research content on therapeutic landscapes in community spaces by means of publication volume analysis, spatial distribution, keyword analysis, co-citation clustering analysis, keyword timeline, and co-occurring word analysis. Based on the research results, the current status and direction of related research are outlined, and the research hotspots and future trends in this field are analyzed. Current research comprises multiple interdisciplinary branches, such as geography, public space, modern medicine, care, horticultural therapy, urban ecology, and more, with theoretical research, caregiving, spatial territories, and research methodology as the main research vectors. It is clear from this study that the current research on community therapeutic landscapes suffers from a lack of coordination between theoretical and practical development, and the related design practice activities are in a vulnerable stage of development. In terms of the population served, specialized research will be one of the directions of development, as there has been a gradual increase in the number of spatial research on the prevention and complementary treatment of various diseases for subdivided groups. At the same time, the research focus in this field has shifted from the physical health of users to their mental health, leading to a trend of public service development with the objective of social health.

**Keywords:** therapeutic landscapes; community open spaces; mapping knowledge domains; visual analysis; CiteSpace





## 1. Introduction

The swift evolution of human society has led to an increasing number of cities, expanding urban dimensions, and a growing urban population. As reported by the United Nations, it is predicted that by 2050, roughly 68% of the global population will reside in urban regions [1]. Although prosperous cities offer many conveniences for people's lives, the high population density, buildings, roads, public facilities, etc., constantly encroach on natural spaces in the city, resulting in the serious degradation of the natural environment in high-density urban areas. In addition, unfulfilled innate biophilia [2] instincts in humans may result in chronic physiological or psychological ailments such as hypertension, heart disease, depression, and anxiety [3,4]. In 1995, the World Health Organization (WHO) introduced the idea of healthy cities [5]. Since then, many countries have adopted this concept as a standard for their urban development initiatives. Under the impact of several global public health and safety incidents, "health" has become a key topic of concern in the current society. World Health Day, which focuses on major public health issues affecting the international community, has gradually shifted its theme from focusing on certain diseases or groups to focusing on the health of all people in order to build an equitable and healthy

world, and the theme for 2023 has been designated as "Health for All". On 21 September 2023, a new political declaration was approved at the United Nations Headquarters during a high-level meeting titled, "Universal Health Coverage: Expanding Our Ambition for Health and Well-being in a Post-COVID World". Specifically, governments have pledged to achieve universal health coverage by 2030.

The health benefits of nature are extensive and significant [6]. Although the positive perception people have of nature is evident, the exact confirmation that the natural environment can impact people's health did not occur until 1984, when the environmental psychologist, Roger Ulrich, pioneered the argument that "integrating natural landscapes into therapeutic environments can aid patients in their recovery" [7]. With the introduction of the term "therapeutic landscape" [8], scholars in several countries have initiated a variety of qualitative and quantitative research as well as practical interventions considering the therapeutic role of the landscape. It was demonstrated that landscaped green spaces have a positive correlation with promoting human attention, mood, and physical and mental health. These areas have the potential to build well-being and promote public health and safety [9].

Community open landscape spaces are frequently contacted by urban residents as a natural environment, and they provide an objective advantage for residents' health. They are not only an affordable means of promoting personal development and health benefits, but also a way to build social well-being. Against the backdrop of a high incidence of public health and safety incidents, on World Cities Day 2020, UN Secretary General António Guterres noted that cities have suffered greatly from the current COVID-19 outbreak and that the value of community has been a particular focus of attention during the outbreak. Therefore, concentrating on the design of community open spaces with therapeutic functions plays a pivotal role in taking care of socially disadvantaged groups and achieving equity in green spaces and sustainable lifestyles [10,11].

In this paper, we conduct a systematic review of the research literature on therapeutic landscapes in community spaces from 2000 onwards. Using CiteSpace, we organized and mapped the knowledge domains pertaining to the topic of "health-landscape-community". This approach allowed for an understanding of the therapeutic landscape research trends in community open spaces, an analysis of current research fields, hot spots, and trends in this topic, and the provision of reference information and guidance for future research.

## 2. Materials and Methods

### 2.1. Analytical Methods

Mapping knowledge domains in scientometrics is a structural representation of the evolution and transformation of scientific knowledge, possessing the dual nature of a "diagram" and a "spectrum" [12,13]. Mapping knowledge domains presents knowledge graphs and genealogies in a visually represented form. CiteSpace is a knowledge visualization software tool that combines the principles of bibliometrics and information visualization [14]. It presents complex relationships such as interactions, evolution, and networks between knowledge units and clusters in an intuitive visual presentation [15]. Through CiteSpace, knowledge mapping allows for a systematic literature review of a particular research field, presenting the research's structure, developmental patterns, and trend detection in graphs and charts that are easy to understand [14,16,17]. Therefore, it can be used to analyze the knowledge structure, evolution patterns, and future directions of the field in an objective and concise manner.

In this paper, CiteSpace 6.2.R1 was used to visualize and analyze the literature data related to community therapeutic landscapes retrieved based on the Web of Science database, and a systematic literature review was conducted on this basis. The current status and trends of research on therapeutic landscapes within community spaces were structured through various analytical methods including spatial distribution, co-occurrence analysis of keywords, co-citation clustering analysis, time zone chart of keywords, and noun term burst detection [18]. This study used CiteSpace knowledge graph visualization to analyze

the literature data in terms of baseline situation, research hotspots, and evolutionary trends, respectively, with the aim of sorting out the current research status and future development trends of therapeutic landscapes related to community environments. In this study, the construction of a knowledge graph (Figure 1) and the analysis and mining of the literature data were carried out in the following steps.

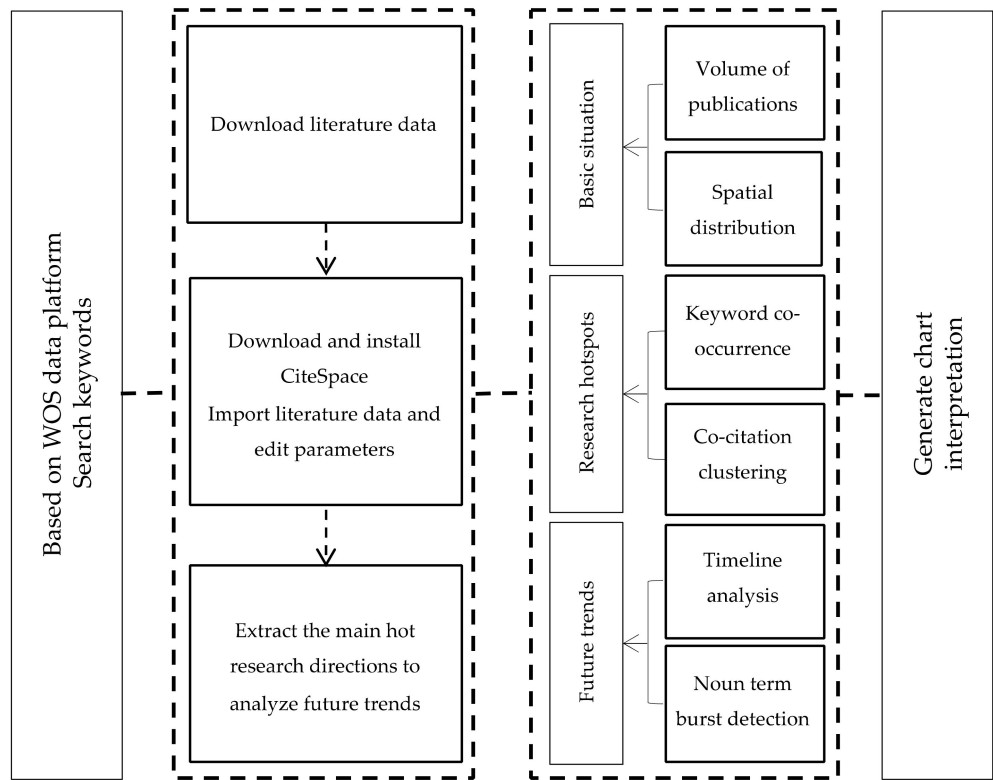

**Figure 1.** Flowchart for building a knowledge graph.

## 2.2. Data Collection

Web of Science (WOS) is a widely utilized, authoritative literature database that includes four independent databases: SCIE, SSCI, CPCI, and A&HCI. These databases encompass the natural sciences, social sciences, humanities, arts, and other multidisciplinary fields, including journals, conferences, reports, and other forms of the literature. This study was conducted in June 2023 and utilized the WOS Core Aggregate Database as a source of literature data to globally analyze research on therapeutic landscapes within community spaces. To ensure comprehensive and reliable literature data, we made multiple comparisons and adjustments to the search strategy. Finally, we initiated the search on the WOS core ensemble database using the themes "S = (therapeutic landscape OR healing garden OR horticultural therapy) AND (community OR residential) AND language = English". Meanwhile, two types of papers, research papers and literature review papers, were chosen to ensure that the literature data could thoroughly depict the research themes and characteristics. Since the volume of the literature published prior to 2000 ranged from one to three articles annually, the dataset was too limited. Therefore, this paper's search range was limited to 2000–2023, resulting in a total of 1254 pieces of the literature. After filtering out conference proceedings, news items, letters, reviews, and irrelevant fields, 538 valid literature pieces remained as the research data. The data collection steps are shown in Figure 2.

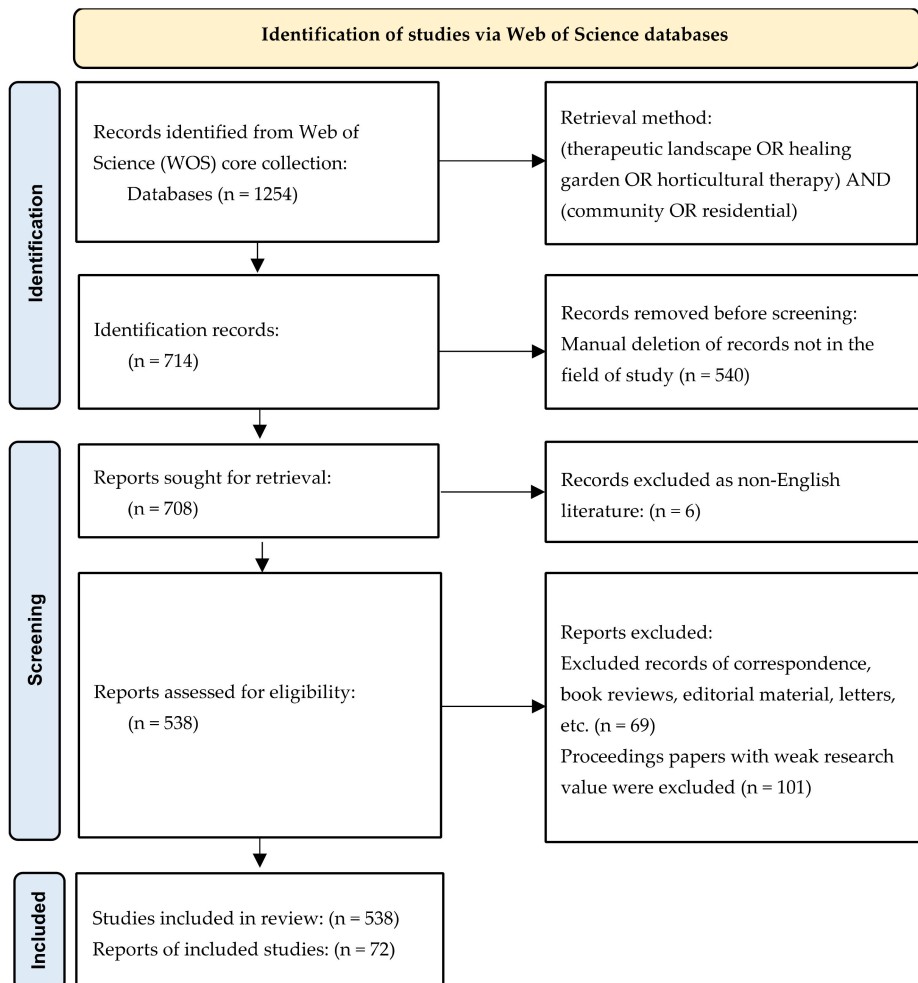

**Figure 2.** The data collection steps.

## 3. Results

### 3.1. An Overview of Basic Research

#### 3.1.1. Number of Papers Published

The statistical data from this study show a positive correlation between the amount of the research literature about therapeutic landscapes in community spaces and time, with an expected increase in the future (Figure 3). According to the analysis of literature data, the research in this field could be divided into three stages: 2000–2012 is the Fluctuating Exploration Phase, with the growth of research in this phase being fluctuating and slow, the highest annual number of articles being 16, and the lowest annual number of articles being 4, which was extremely unstable; 2013–2016 is the Fluctuating Development Phase, during which the fluctuation still had its ups and downs during the four-year period, but the number of articles increased significantly, and the number of articles was more than 20 during the period of three years; and after 2017, is the Rapid Growth Phase, which had a rapid growth trend of between 30 and 80 annual articles. After 2017, there was a rapid growth phase, in which the annual number of articles between 30 and 80 showed a rapid growth trend, and although there were fluctuations during this period, the number of articles was still considerable. In summary, it is expected that there will be a continued increase in the number of articles in this field in the future.

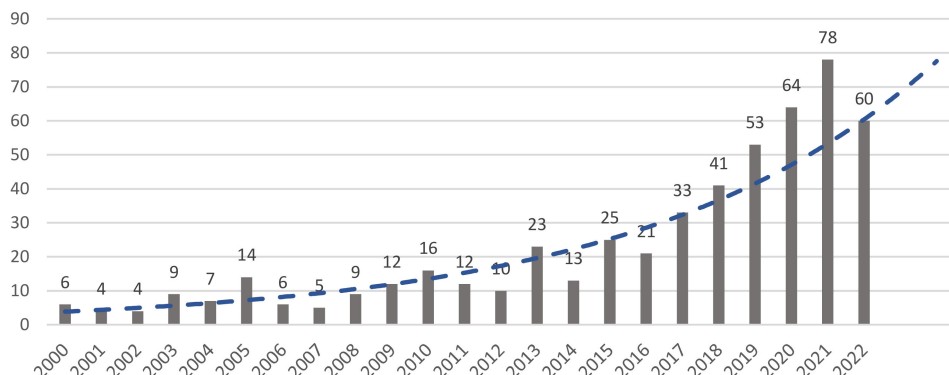

**Figure 3.** Literature Volume and Trend Statistics Related to Rehabilitation Landscapes, 2000–2022 (Statistical Time 4 June 2023). Since the literature data of 2023 are not complete, the statistics and analysis of the number of publications are as of December 2022.

The specific research characteristics of each phase of research related to therapeutic landscapes within community spaces are listed below:

- Fluctuating Exploration Phase (2000–2012): Studies from this period fall into two main categories. The first category is based on the need for human health services, which has driven the introduction of initiatives, concepts related to community services, and programs for community care services and health services. For example, Cattell, Vicky, and others argued that social interaction in public spaces plays an important role in improving people's mental state and maintaining a sense of community and that people gain a sense of well-being through public spaces in their neighborhoods [19]. Wolfe, Mary K., and others, on the other hand, found through their study that urban residential neighborhoods with well-maintained vegetation had a reduced incidence of certain crime types [20]. The second category comprises quantitative research on the effects of natural factors and horticultural activities on human health, again providing data to support the idea that the natural environment and natural activities affect human health. For example, Voeks and colleagues found that being female, older, less illiterate, more educated, and more knowledgeable about medicinal herbs were positively associated with a greater likelihood of having a positive health outcome [21]. Hale-James and others argued that aesthetic relationships are fundamental connections between people and that the stimulation of the senses through horticultural activities, as well as the learning, communication, recognition, and social relationships that result from the horticultural process, contribute to health promotion [22].
- Fluctuating Development Phase (2013–2016): The field of study entered a phase of fluctuating development during this period due to the continued interest in therapeutic landscapes and community open spaces, as well as the emphasis on the health of the population. In this phase, research began to focus on the health needs of "people" in the direction of population segmentation and assistance in alleviating disease; for example, in terms of population, it began to focus on ethnic minorities, refugees, orphans, patients, and people of all ages. In terms of assisting in the alleviation of disease, the main focus was on cancer, Alzheimer's disease, wartime trauma, and other physical and mental illnesses. Extensive quantitative research has also been conducted on the effects on human health of elements of the natural environment, represented by flora and fauna, outdoor facilities, music, and behavioral styles, as well as elements of the artificial environment and human behavior. For example, in the case of plants, Koga, Kazuko, and others found that people experience an unconscious calming response when they touch plants [23]. In terms of behavioral approaches, Doughty argued that social group dynamics, such as walking together, are essential components of many therapeutic landscapes [24]. During this period, research in this area started to have an impact on government policy concerning "well-

being". Dinnie et al. suggested a more in-depth exploration of social relationships and social health in relation to green spaces and their management [25]. Additionally, the integration of traditional gardens with rehabilitated landscapes emerged during that time.

- Rapid Growth Phase (2017–present): This phase of the study on therapeutic land-scapes in community open spaces is expanding to multiple disciplines, enabling a more comprehensive interdisciplinary discourse. Research on the characteristics, psychology, behavior, and diseases of diverse populations has become increasingly targeted. Notably, research on the elderly has gained prominence due to the global aging trend [26,27]. Research on the impacts of different natural factors on health and the effects of physical activity in natural settings on health have been investigated more extensively and specifically [28–30]. Quantitative studies on the impact of nature on human health have generated objective experimental data from various multidisciplinary fields [31,32]. These findings can provide more valuable evidence regarding the health benefits of natural environments. On the other hand, while the benefits of nature for human health are well known, healing landscapes are a means of promoting health, not a way of ensuring it. Different forms of therapeutic landscapes can have different effects in different settings; for example, residents of some poor communities do not find weakly attractive, inadequately maintained therapeutic landscapes to be physically and spiritually healing [33].

### 3.1.2. Spatial Distribution

In terms of the cyberspatial distribution of the established literature data, the countries with a high number of publications in the field of community-based therapeutic landscape research are mainly Europe and the North America, represented by the United States, England, Canada, and Italy, Asia, represented by China and South Korea, and Australia also has a notable presence. In this paper, one year was used as a time slice (2000–2023), and each country was mapped as a node for visualization and analysis (Figure 4). Among them, the United States and England had far more publications than other countries due to their earlier research in the field of therapeutic landscapes. Australia and China, although late in entering the research field, also showed vigorous development.

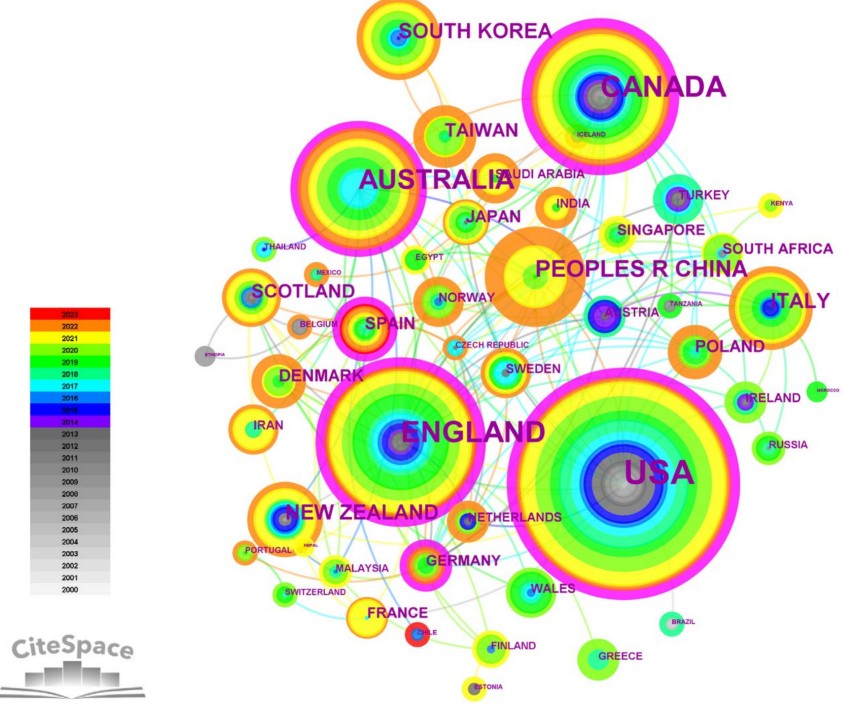

**Figure 4.** Visualization of the country collaboration network from 2000 to 2023.

In CiteSpace, if the betweenness centrality of a node is high, it means that the node plays a pivotal role in that subject area and also plays a vital role in connecting other nodes [34]. For the study of therapeutic landscapes within community spaces, national mediated centrality data (Table 1) were tabulated and categorized into two categories. The first category is represented by England, the United States, Canada, Australia, Germany, and Spain, all of which have a betweenness centrality > 0.1, indicating a high level of importance and influence in this area of study, and they are positioned at the forefront of research in this field. Although some countries have fewer publications, their depth and influence can be significant. For example, even though England started at the same time as the United States in this field, their number of publications is much lower. However, England's betweenness centrality (0.27) is still in first place, indicating that the English literature on community-based therapeutic landscapes is more influential and important than that of other countries. The second category includes Italy, China, Scotland, Japan, and Poland, each having a betweenness centrality value of <0.1 and >0.01. Although these countries have a higher volume of literature in the area of community-based therapeutic landscape research than others, the quality of their output literature is variable, lacking in both breadth and depth compared to the first group of countries. According to the statistics on the number of articles, among the countries with a high number of articles, South Korea has a high number of articles in the field of community therapeutic landscapes, but its betweenness centrality is "0", and although it has a relatively high number of articles, it does not have an impact on other countries.

**Table 1.** Statistics on the number of publications and betweenness centrality of research papers on community therapeutic landscapes.

| Country | Frequency | BC [1] | Time | Country | Frequency | BC [1] | Time |
|---|---|---|---|---|---|---|---|
| United States | 149 | 0.23 | 2000 | New Zealand | 18 | 0.01 | 2003 |
| England | 84 | 0.26 | 2000 | Scotland | 13 | 0.06 | 2002 |
| Canada | 67 | 0.2 | 2002 | Poland | 12 | 0.04 | 2008 |
| Australia | 49 | 0.12 | 2009 | Spain | 10 | 0.1 | 2009 |
| China | 33 | 0.06 | 2009 | Japan | 9 | 0.05 | 2012 |
| South Korea | 20 | 0 | 2010 | Denmark | 9 | 0.03 | 2017 |
| Italy | 20 | 0.07 | 2013 | Germany | 7 | 0.18 | 2013 |

[1] Betweenness centrality (BC): A node metric measuring how likely an arbitrary shortest path in a network is to go through the node, which shows the node-to-node connectivity contribution within a network [35].

This paper concludes by combining studies from countries with a high betweenness centrality and number of publications that have relevant results in this field. The following conclusions were drawn.

1. In Europe and North America, led by the United States and England, research and practical activities related to the integration of natural landscapes into community spaces such as medical resources and social well-being have been relatively prominent and far-reaching, radiating to the "attention to various groups of people, prevention of diseases, and methods of design".
2. Asian countries such as China, Korea, and Japan are more likely to study the connections between plant characteristics and therapeutic landscapes in terms of horticultural therapies, and there has been more research and attention related to the characteristics of elderly users.
3. With the global emphasis on public health and the expectation of a high quality of life, scholars from various nations are increasingly studying therapeutic landscapes in community spaces. These scholars are examining strategies, services, and designs of therapeutic landscapes in community spaces from a variety of perspectives.

Studies in Europe and North America have argued that personalized medicine, based on innovative scientific and social infrastructures and high-quality healthcare facilities, should be equitably distributed in urban communities [35,36] to achieve reciprocity between society and individuals. Among them, reciprocal opportunities are the basis for the



development of healthy communities with heterogeneous groups [37]. Brown, Steven D., and others developed the concept of "vitality" to describe the contribution of the environment to the "feeling of being alive" [38]. Cleary, Anne et al. argue that connecting with nature can help promote well-being and can make a significant contribution to public health [39]. Engineer, Altaf et al. developed an environmental framework for social well-being in seven components: sleep, resilience, environment, exercise, relationships, spirituality, and nutrition [40]. Cravey, A.J., and others argued that socio-spatial knowledge networks (SSKNs) can be used for healthcare prevention strategies and provide design information [41]. Cattell, Vicky et al., proposed a policy approach to the debate about public spaces in relation to well-being and society, arguing that public spaces with therapeutic properties should be widely recognized [19].

In Asian nations, particularly China, Japan, and Korea, horticultural therapy has been researched quantitatively, revealing its capacity to enhance and sustain physical health while encouraging overall mental and emotional well-being via community gardening [26,42,43]. Furthermore, research indicates that individuals may experience an unconscious calming response when in contact with plants [23]. According to Lee, Juyoung, traditional gardens that contain historical and cultural elements also have the ability to reduce negative emotions and function for the well-being of urban residents [44]. Yu, Shiwang et al. showed that social, leisure, and functional activities of older adults in an aging residential community were associated with environmental noise, amenities, and green space [45].

Other nations with a mediator centrality below 0.01 have also demonstrated dynamism in this field of study. For example, in Melbourne, Australia, a settlement environment for young refugees was created. Furthermore, Thai scholars have proposed therapeutic strategies and interventions for the emotional therapeutic landscapes and special cultural needs of female breast cancer patients and the establishment of hospice gardens for the elderly [46,47]. These research studies primarily focus on therapeutic landscapes related to the analysis of the needs of minorities, disadvantaged communities, and other subgroups of user groups. They strive to promote the innovation of user-centered spatial environment creation from multiple perspectives. At the same time, some scholars have paid attention to the possibility of developing therapeutic landscapes in community spaces from the perspective of ethnic identity. To illustrate, researchers have focused on local plants by integrating ethnopharmacology-related content and expanding their functions to edibles, teas, and so on, in order to explore the possible medicinal (health) value, recreational value, and ethnocultural value of landscape plants in the spatial community [48,49]. In addition, some scholars have studied the role of religious emotional sustenance in community therapeutic landscapes from the perspective of religious belief [50,51]. Scholars from different countries have provided relevant research on how design intervenes in the innovation of therapeutic landscapes in community spaces from different perspectives. Among them, Iranian scholars have investigated the role of artificial canals, namely canals, in providing urban health and have proposed a framework for the therapeutic effects of blue-green spaces from a design perspective. To sum up, the participation of more and more researchers has prompted the research on therapeutic landscapes in community spaces to break through traditional research frameworks and methods, trying to diverge from multiple perspectives such as subdividing groups and multidisciplinary integration, and constantly trying to integrate. In order to promote richer and more diversified research among them, there is still a lot of room for innovative research on design interventions in community therapeutic landscapes.

### 3.2. Research Frontiers and Hotspots

### 3.2.1. Co-Occurrence of Keywords

Keywords chosen from the title, abstract, and text of the paper are a distillation of the content of the document. This paper starts by analyzing high-frequency keywords and keyword co-occurrences to offer an overview of present trends in development and research hotspots in the field. Among them, high-frequency keywords mean the keywords

with the highest frequency in the analyzed literature data. A high-frequency keyword analysis method is a method used to extract the high and low distributions of keywords or subject words that can express the core content of the literature on the information of literature data and to analyze the current popular research fields and related trends from the visualized data. The simultaneous occurrence of multiple keywords is called keyword co-occurrence, which is a two-by-two count of a set of words in terms of the number of times they appear in the same set of documents, and their proximity is measured by the number of such co-occurrences [12].

In this study, we obtained the co-occurrence mapping of high-frequency keywords related to community therapeutic landscapes by using CiteSpace software, which was run on the WOS database. The results are illustrated in Figure 5. In the graph, each node depicts a keyword, with larger nodes indicating higher frequency. The color of the graph transitions from purple to yellow, with the nodes closer to yellow indicating frequent appearances in the recent literature. In the current international research on therapeutic landscapes in community open spaces, in addition to the keywords, "therapeutic landscapes" and "community" searched for in this paper, high-frequency keywords with a betweenness centrality 1>, such as "geography", "health", and "horticultural therapy", as well as keywords with a betweenness centrality close to 0.1, were also found in this paper. The keywords "mental health" and "place", which have a betweenness centrality close to 0.1, are also hot topics in this research area (Table 2).

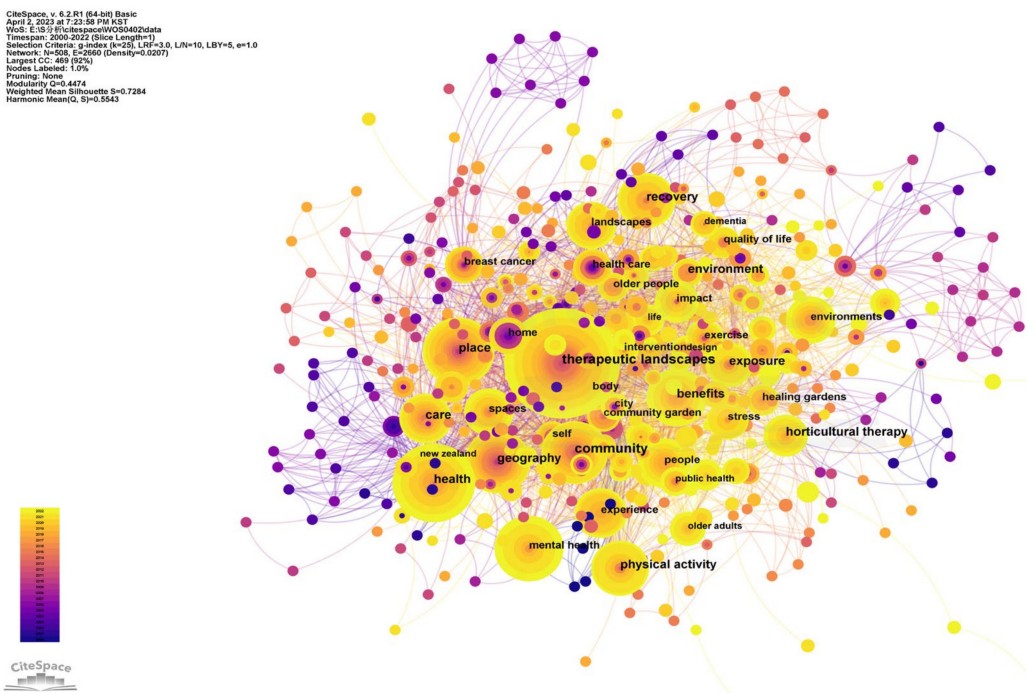

**Figure 5.** Keyword co-occurrence network mapping.

**Table 2.** Community treatment landscape of high-frequency word statistics, 2000–2023.

| High-Frequency Keyword | Frequency | BC | Year |
|---|---|---|---|
| Therapeutic landscapes | 159 | 0.31 | 2001 |
| Community | 55 | 0.18 | 2005 |
| Care | 65 | 0.14 | 2001 |
| Geography | 56 | 0.12 | 2003 |
| Health | 102 | 0.12 | 2004 |
| Horticultural therapy | 38 | 0.11 | 2001 |
| Place | 63 | 0.09 | 2001 |
| Mental health | 53 | 0.09 | 2000 |
| Benefits | 52 | 0.09 | 2002 |
| Breast cancer | 13 | 0.09 | 2006 |

### 3.2.2. Co-Citation Clustering

Citation analysis is the basis of literature co-citation analysis, with scholars citing previous research results in their papers and listing them in the form of references in their studies, thus generating new knowledge, and this constant citation between the scientific literature illustrates the accumulation, continuity, inheritance, as well as disciplinary crossover and penetration of scientific knowledge [12]. Co-cited literature can form multiple reference clusters. Each reference cluster represents a common research topic. The process of mining multiple co-citation relationships through the literature network is known as co-citation analysis of the literature [52]. Analyzing the highly cited literature, which is often the basis for the development of a theme, reveals the progress and direction of research on "community therapeutic landscapes" [13].

The highly cited literature in this research was acquired via CiteSpace utilizing the log likelihood ratio (LLR) algorithm to generate the literature co-citation cluster mapping (Figure 6). The results showed Q = 0.9069 and S = 0.9684, indicating that the clustering network had significant modularity as well as high homogeneity and a significant clustering effect, which could be effectively analyzed. In order to maintain the clarity of the clusters, only the nine clusters with a high number of citations and strong homogeneity were highlighted and sorted, and nine co-cited clusters were finally obtained after screening (Table 3). By reading in detail the frequently cited references in the nine clusters in detail and combining this with the keyword analysis in the previous section, the research hotspots and frontiers of community therapeutic landscapes were deduced.

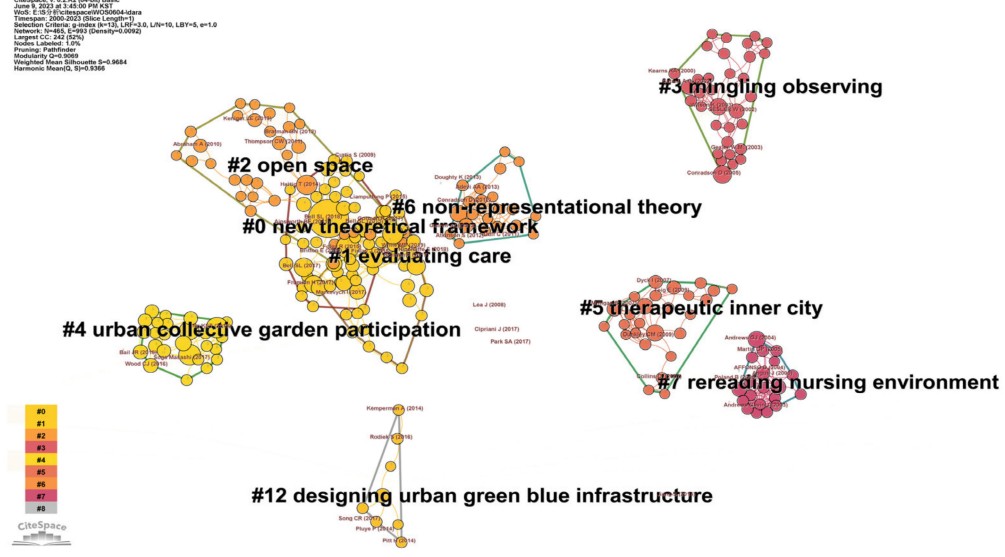

**Figure 6.** Co-citation and cluster analysis of the literature related to therapeutic landscapes and community open spaces.

**Table 3.** Literature co-citation clustering table, 2000–2023.

| Cluster ID | Size | Silhouette | LLR [1] | Average Year |
|:---:|:---:|:---:|:---:|:---:|
| #0 | 38 | 0.938 | New theoretical framework | 2016 |
| #1 | 36 | 0.953 | Evaluating care | 2017 |
| #2 | 34 | 0.96 | Open space | 2011 |
| #3 | 33 | 0.994 | Mingling observations | 2002 |
| #4 | 26 | 0.98 | Urban collective garden participation | 2018 |
| #5 | 26 | 0.969 | Therapeutic inner city | 2007 |
| #6 | 21 | 0.976 | Non-representational theory | 2011 |
| #7 | 20 | 1 | Rereading nursing environment | 2003 |
| #12 | 8 | 0.974 | Designing urban green blue infrastructure | 2015 |

[1] Log likelihood ratio (LLR): This is a significant statistical test used for determining whether two or more samples originate from the same probability distribution.

This study summarizes the discovery of four main lines of research and lists the core elements as follows.

1. Theoretical-non-representational theory—A new theoretical framework.

Therapeutic landscape theory, an extension of environmental psychology, cultural (health) geography, medicine, and environmentalism, has gained significant attention and recognition across society since its inception. The concept of therapeutic landscapes acting as cultural landscapes in healthcare proves once again that people are closely linked to their environment, society, and space [8]. As the concept of therapeutic landscapes has been studied intensively, non-representational theories have come to the fore. Non-representational approaches are not strictly speaking theories, but rather ways of comprehending the world and are thus widely employed in the practical realm. Since the focus of non-representational theory has shifted from the study of the intrinsic meaning of things to the subtle and unintentional manifestations and practices of life, it involves making connections to human behavior and the senses [53]. Happiness, viewed as a pursuit for a better life, and its potential impact on health are not only superficial but may also be influenced by environmental factors. Therefore, exploring established concepts and theories within health-related professions and fields could form the basis for future research [54]. The relationship between human behavior, represented by walking, and well-being and health has been theorized as "therapeutic mobility" and can also be explored through activity, connection, and the context that walking improves physical health and mental health [55]. When therapeutic landscape theory and non-representational theory are combined, a new theoretical framework on the field continues to emerge. For example, Sarah L. Bella et al., examined the concept of therapeutic landscapes, mapped key applications of the concept in its core physical, social, spiritual, and symbolic dimensions, and applied the concept to the marginalized in society as a key challenge and suggested that beneficiaries are diverse and differentiated in terms of their health or healing as it develops over time. Diverse understandings of current landscapes, assemblages, and practices across a range of approaches is another key challenge, which Sarah L. et al., argue should be combined with practical physical and psychological measures to facilitate public empathy [9]. Ronan Foley and Thomas Kistemann developed the concept of "blue space" through their research on therapeutic landscapes and the relationship between the environment, health, and well-being, defining blue space as "health-promoting places and spaces where water is at the center of a range of environments with identifiable potential to promote human well-being" [56]. Jessica Finlay et al. extended the concept of therapeutic landscapes by analyzing data from interviews with older adults aged 65–86 and argued that blue spaces should be more prominent in health policy and urban planning for older adults [57].

2. Care—Rereading the nursing environment and evaluating care.

Care can refer to either a limited sense of personal care, where an individual is unable to fulfill their daily physical needs and must depend on others to assist them, or a broader sense of environmental or social care, where nursing activities are carried out within a social sphere [58]. Care in research related to therapeutic landscapes generally refers to care in a broad sense. The extension of healing into the landscape of non-medical environments and the utilization of the landscape's healing power to create "care" in social spaces is one of the most important ways in which therapeutic landscapes can promote social justice. This study found that several scholars have reinterpreted care environments after 2003 and found that care environments of a physical nature should be places with a variety of qualities, care experiences, and places where emotional attachment can be generated; embedding autonomy, quality of life, care, and treatment in this context is a productive and meaningful improvement [59] and therefore incorporating the concept of the therapeutic landscape into aspects of treatment and care can provide direction for the development of healthcare and care environments [60]. The care setting is not limited to the therapeutic environment, but can also be social spaces such as the home and the community, with the overlap of the home and the healthcare space being favored by those being cared for [61].

In the context of the increasing number of care environments, when care environments are shifted to spaces characterized by "local culture", the spatial environment of human settlements in public health discourse is more amenable to health [62], thus promoting social care and social justice. In therapeutic landscapes, caregiving agriculture refers to the use of agricultural practices to promote health, as a way of realizing the integration of social and educational services and as a new way of promoting healing and recovery [63,64]. Thus, horticultural activities, animal care, social communication, and other nature-based interactions are increasingly acknowledged by researchers as health-promoting forms of social care with considerable potential for traumatized psychotherapeutic aspects [65]. In addition to plants, animals play a crucial role in promoting health as an integral aspect of care farming. Therefore, constructing healing spaces should not solely be based on human needs and desires, but also on the position of animals [66]. Sensory experiences are particularly important in therapeutic spaces in terms of healing [67]; for example, Richard Gorman argues that scent positively influences practice and engagement with places and that scent can transcend the audiovisual and be more fully integrated into the therapeutic landscape [68].

3.  Place—Geography—Community—Therapeutic inner city—Open space—Designing urban green blue infrastructure—Urban collective garden participation.

In the theory of "sense of place", the subjective nature of human perception of place and the significance of positive subjective feelings in spreading well-being and creating value [69,70] have led to a wide range of research on place since 2001. Health geography is one of the contexts in which the theory of therapeutic landscapes was formed, which argues about landscapes mainly in cultural, theoretical, and medical dimensions [71]. As space has been increasingly studied as a place of meaning, health geography has also approached the study from the perspective of a "place-environment" and recognized the close relationship between therapeutic landscapes and spatial geography [72]. Familiar environments as opposed to unfamiliar environments create place attachment, and people are more inclined to stay in places where they feel safe, comfortable, and have a sense of belonging [73]. The character of therapeutic landscapes changes in response to different environmental changes due to the variability in natural environments, economic circumstances, cultural beliefs, and ethnic identities. Geographies of well-being, for example, are inspired by indigenous cultures and geographies of health, emphasizing the cultural specificity of indigenous landscapes and the need for therapeutic landscapes to change based on changes in the environment of the location [74]. As the study of therapeutic landscapes in terms of place and geography progresses, communities are characterized by a sense of place as well as unique cultural and health geographies, making them spatial environments that can be further developed. In addition, communities are important in terms of a sense of belonging, social interaction, liberation from the margins of society, and solidarity against oppression. It has been found that therapeutic landscapes formed by people as subjects engaging and communicating with the natural environment through interactions in taskscapes [75] and retreats [76] can greatly contribute to the development of community-based urban environmental justice [77]. Currently, because of the uneven development of research on nature for health, the therapeutic role of the natural environment should be actively and critically embraced [78,79], the formation of healing cities should be reached from the perspective of therapeutic landscapes in communities, and healing city research should be extended to a wide range of open spaces. By triggering interactions with cultural, geographic, and economic groups in open spaces, landscapes that promote high-quality interactions between people and nature [80] are important for achieving social integration, social participation, social support, and social security, and thus, social well-being [81]. As people age, their mobility tends to decline, along with a reduction in both their space and the range of activities they are able to undertake. At present, the availability of trees and grass, along with perceptions of greenery, are the primary factors impacting social connections among neighbors in community spaces [82]. In addition, there is a greater preference for natural, aesthetically pleasing, and diverse landscaping, as well as spatial

environments featuring good infrastructure and easy accessibility [83,84]. In addition, horticultural therapy in urban collective garden participation is also prominent in this regard. Therefore, the integration of horticultural therapy into public gardening can not only improve physical, mental, and social health [85]. In the long run through the feasibility of horticultural therapy interventions [86], it can also alleviate and prevent various health problems currently faced by society [87,88], thereby improving social well-being [26].

4.  Mingling observations.

In the study of community open space, scholars usually use mixed methods represented by observing the site, semi-structured interviews, qualitative research, and quantitative experiments to study open space and reach a consensus on the unique healing characteristics that a community open space possesses, such as comfort, inclusiveness, ethnicity, and culture [19]. For example, Kathleen Wilson researched the relationship between place and health in specific cultural contexts through interviews, arguing that culture is an important part of open spaces in everyday life [89]. David Conradson, on the other hand, argued that the therapeutic landscape is best approached as a relational outcome, as a series of exchange scenarios that pass between the individual and his or her wider social environment [90]. In addition, Christine Milligan and colleagues conducted a study on the health impacts of public gardening activities, using observation, testing, and semi-structured interviews. Their findings suggested that gardening activities can lead to a sense of accomplishment, satisfaction, and aesthetic pleasure and that public gardening activities promote inclusivity by creating shared spaces [61]. Tanja Schmidt and her colleagues conducted a quantitative study of community open spaces using observation and interviews. Their findings concluded that social interaction is a pivotal factor in the utilization of community open spaces amongst older individuals [91].

*3.3. Future Evolutionary Trends*

3.3.1. Keyword Timeline

A timeline map is a visualization map constructed on the basis of keyword co-occurrence according to the year of keyword appearance, which can intuitively reflect the hot keywords of therapeutic landscapes in community space in each year of the research process, according to which we can further understand the stage of development and developmental trends of the research, and on the basis of which we can make a certain degree of predictions on the future direction of the research [16].

As shown in Figure 7, before 2000 was the beginning stage of therapeutic landscape research in community spaces. In 2000–2015, quality of life became a hot topic in the field of therapeutic landscapes and combined with healing gardens, horticultural therapy, green space, and healing environments under the joint research, the research in the field of therapeutic landscapes in community spaces was promoted. There was limited research on therapeutic landscapes in community spaces from 2015 to 2019, with a majority of studies focused on traditional topics. A few studies explored nature-based interventions, civic ecology, modern medicine, space and place, with horticultural therapies emerging as a popular research topic in 2018–2019. The frequent outbreaks of public health and wellness events from 2019 to 2023 have brought community values and wellness to the forefront of people's minds, prompting research on healing gardens, horticultural therapies, and more to remain able to maintain a considerable amount of fervor. Researchers' attention to nature-based interventions such as community gardens, green spaces, and quality of life is gradually increasing, and the soundscape represented by "music" has become a new research hotspot at this stage. In addition, the relationship between therapeutic landscapes and quality of life has become the most important research direction for researchers through literature data.

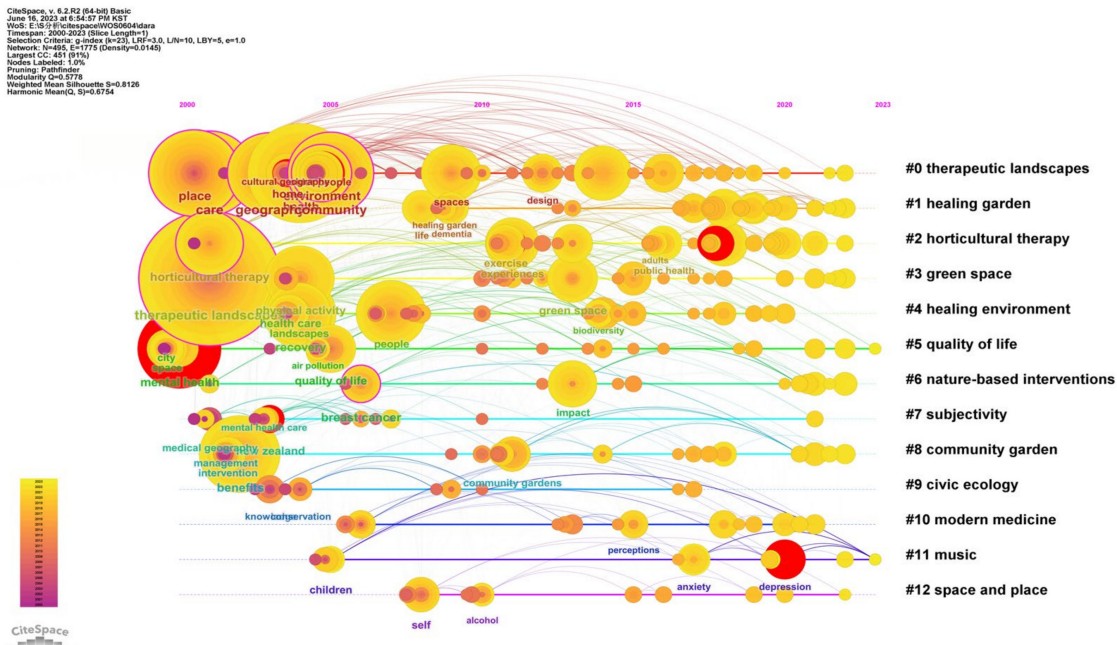

**Figure 7.** Spatial timeline map analysis of community therapeutic landscapes.

### 3.3.2. Noun Term Burst Detection

Burst strength represents the prominence rate of keywords, which can reflect the more influential research areas in a period of time, representing the research hotspots and scientific development research trends in a specific time. A total of 84 burst noun terms were obtained through CiteSpace keyword bursting and burst time visualization. Table 4 shows the 20 most frequent groups of terms.

**Table 4.** Top 20 high-frequency terms and their burst time.

| Keyword [1] | Year [2] | Strength [3] | Begin [4] | End [5] | 2000–2023 |
|---|---|---|---|---|---|
| Medical geography | 2001 | 2.45 | 2001 | 2006 | |
| Home | 2004 | 4.97 | 2004 | 2010 | |
| Self | 2008 | 1.92 | 2008 | 2010 | |
| Geography | 2003 | 2.62 | 2012 | 2016 | |
| Quality of life | 2005 | 2.79 | 2014 | 2016 | |
| Older people | 2005 | 4.13 | 2016 | 2018 | |
| Environment | 2005 | 3.97 | 2016 | 2017 | |
| Community gardens | 2011 | 2.2 | 2017 | 2018 | |
| Experiences | 2011 | 3.37 | 2018 | 2019 | |
| Landscapes | 2004 | 2.97 | 2019 | 2020 | |
| Framework | 2019 | 2.41 | 2019 | 2020 | |
| Therapy | 2019 | 1.93 | 2019 | 2020 | |
| Depression | 2020 | 3.61 | 2020 | 2023 | |
| Stress | 2008 | 2.31 | 2020 | 2021 | |
| Walking | 2020 | 2.31 | 2020 | 2023 | |
| Ecosystem services | 2020 | 2.03 | 2020 | 2021 | |
| Impact | 2013 | 1.83 | 2020 | 2023 | |
| Mental health | 2000 | 3.37 | 2021 | 2023 | |
| Mortality | 2021 | 2.07 | 2021 | 2023 | |
| Green space | 2013 | 2.05 | 2021 | 2023 | |

[1] Keyword: represents explosive noun words; [2] Year: indicates the year when the keyword begins to appear; [3] Strength: indicates the strength of the outbreak; [4] Begin: represents the beginning year of the outbreak of the noun term; [5] End: indicates the end year of the outbreak; The blue lines show when the keywords started and the red lines show the durations of the bursts.

Research topics related to the impact of landscape space on health have been popular among scholars in various countries during the period 2000–2023. Since this field of research was first extended from the fields of environmental psychology, health geography, and medicine, with a focus on the human environment, "medical geography, home, and self" became the first bursts of keywords to become prominent in the years 2001–2010. At this stage, research in the field of medical geography was not only focused on medical environments, but also on residential environments in terms of burst strength and duration, and on the basis of this, on the needs of human beings for their own environments. Since 2012, geography has been at the center of therapeutic landscape research theory for the past five years. During the period when geography was a popular subject, research related to the living environment, the living experience, and the living population iterated with each other on a yearly basis and became a popular research topic, which promoted the development of the research field from multiple perspectives, of which the topic of "elderly" had the most burst strength. Until 2019, the research on the healing nature of landscapes has gradually formed a variety of frameworks, while the healing and facilitating effects of landscapes on psychological aspects has received the attention of more scholars and gradually become one of the main directions in the field of therapeutic landscape research, demonstrating the continuous development and evolution of this research field from the macro to micro levels, from theory to practice, and from the physical environment to the spiritual environment. Mental health research has always existed throughout the entire therapeutic landscape research field, since 2000, and therapeutic landscape-related mental health research was always of high concern but did not form bursts until 2020, in the world of public health and safety outbreaks, leading people to produce "stress, depression" and other emotions, which advanced to 2021 with "mental health" as a keyword for the formation of the strong bursts, along with community therapeutic landscape research into the "focus on mental health-oriented urban green space landscape ecological service". In addition, attention to the mortality rate in the context of population aging is also rising and has become a new research hotspot.

## 4. Discussion and Prospects

Community open space as one of the carriers of urban landscape space, how to use community open space to maximize the possibility of landscape space unique healing, to achieve the role of promoting physical, mental, and social health, and thus improve social well-being, is the main goal of the current community therapeutic landscape-related research, and the same is also for the development of today's social life, the political orientation and the development of the system of attention to content.

Throughout this literature research collation, since 2000, therapeutic landscapes in community open spaces research gradually formed a scale. This paper is based on the community therapeutic landscape of the research objectives, through CiteSpace scientific knowledge mapping visualization and analysis which found that, at present, from the perspective of the amount of articles issued, the current research in this field has entered a stage of rapid growth, and the research on medicine, geography, design, behavior, sociology, and so on between the multidisciplinary discussions is more rich as the most significant feature of this stage. From the results of the visualization study of the spatial distribution of the relevant literature data, Europe and the United States focus on qualitative and quantitative research, while Asian countries are more focused on practical research, and due to the earlier involvement of Europe and the United States in this research field, the impact of the relevant research in Europe and the United States and the quality of the literature is higher than that of the countries. The hotspots of research in this field developed to the present are mainly focused on geography, horticultural therapy, mental health, and so on. It can be seen that the focus of research has shifted from the pure pursuit of quality of life and passive perception of the healing effect of the landscape, to the active experience and perception aspects in the landscape space, but also from the initial pursuit of physiological health research, to focus on the user's multi-level health needs in terms of mental health

and spiritual health research. An examination of research techniques regarding therapeutic landscapes in community spaces indicates that traditional mixed research methods, including behavioral observation, social science statistics, and quantitative testing and analysis, are still more frequently employed. The research encompasses three primary research directions focused on theory, care, and place. Specifically, new theories and design frameworks extended from theories related to therapeutic landscapes and non-representational theories; extending from the narrow definition of caring care to the environment, deriving care activities in non-medical environments, and building on this foundation, such as landscape character extraction and landscape evaluation; as well as place-based research on open space, therapeutic infrastructure, and public engagement.

At the same time, this paper examines the trends in the research field of community therapeutic landscapes through the analysis and organization of CiteSpace's scientific knowledge map. The examination will focus on three areas: application practice, service targets, and service direction. In the application practice, the current community therapeutic landscape research still exists in regional development of the situation and is not coordinated, although the existing research presents a large number of quantitative and qualitative research content on the design of the practice but is still in a weak position, which provides more possibilities for relevant researchers to intervene in the practice of the activities, to promote the community therapeutic landscape continues to develop and innovate. However, studies focusing on subdivided groups tend to primarily focus on minority groups, while research on community-based therapeutic environments for overall health remains prevalent. Therefore, subdivision should be considered based on the prevailing research. In terms of service targets, research on special groups represented by the elderly, as well as spatial research on the prevention and complementary treatment of various types of diseases, has shown an increasing trend, which provides a theoretical basis for research based on the needs of a particular subgroup, and at the same time breaks the concept of the traditional community therapeutic landscape and promotes its development from the perspective of the user's needs more often. In the service direction, with the gradual improvement in medical conditions, the current spiritual needs of people are on the rise, reflected in the community open space requirements in addition to facilities to meet the needs of physiological health, should be more spiritual aspects of service to achieve the purpose of promoting mental health. Similarly, horticultural therapy, as a current means of natural active healing, should be optimized in terms of service and jointly promote the development of social public services. The service aspect can be integrated with service design and other disciplines to form a therapeutic landscape service design system within the community space. In summary, research and practice related to therapeutic landscapes in community open spaces is still at an early stage and has significant potential for growth. Currently, a lack of balance and insufficient volume of publications, spatial distribution, and theoretical research, practical inquiry, and research methodology exists. Consequently, there is a need to integrate various spaces, disciplines, and approaches to explore the development of therapeutic landscapes within communities.

**Author Contributions:** Conceptualization, Y.H.; methodology, Y.H.; software, Y.H.; validation, Y.H. and Y.L.; formal analysis, Y.L.; investigation, Y.H.; resources, Y.H.; data curation, Y.H.; writing—original draft preparation, Y.H.; writing—review and editing, Y.H. and Y.L.; visualization, Y.H.; supervision, Y.H. All authors have read and agreed to the published version of the manuscript.

**Funding:** This research received no external funding.

**Institutional Review Board Statement:** Not applicable.

**Informed Consent Statement:** Not applicable.

**Data Availability Statement:** The articles were obtained from Web of Science (https://www.webofscience.com/, accessed on 4 June 2023).

**Conflicts of Interest:** The authors declare no conflict of interest.

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
