# Peer review of "Scientific Knowledge Map Study of Therapeutic Landscapes and Community Open Spaces: Visual Analysis with CiteSpace"

_sustainability, doi:10.3390/su152015066_

Round 1

Reviewer 1 Report

Scientific Knowledge Map Study of Therapeutic Landscapes and Community Open Spaces: Visual Analysis with Citespace

 The concept is very interesting. The paper is very comprehensive with background information well-covered and significant added knowledge.

Few comments for authors to better improve the manuscript and make it ready for publishing is:

Abstract: please add more details concerning the review process.

Method: please specify the type of review process, is it systemic? authors can use the prisma method and chart for this purpose.

Authors can consider deleting table 1 as it includes repeated information from the cite score website regarding the definitions of the terms used.

Missing figure caption for the data collection step. This chart can be developed based on the prisma chart as previously suggested

The result section is written such that it integrates the discussion section within its folding and compare to previous studies. Nevertheless, I would highly recommend separating the two sections to clearly pinpoint the outcome of this study, then, a separate discussion section that argues these findings and compares them to previous studies.

Result subsections can include: temporal analysis, spatial distribution…etc.

Can the features of therapeutic landscape varies based on the variation in the surrounding context and natural environment?

It is much appreciated that authors have attempted to present a temporal tracking of the increase of number of related publications, and their content but more work is also needed. Thus, please revise the result section in the sense of deducting information and reading what is behind the image simply generated using a software program. In this case, authors’ insightful analysis and critical thinking are required to add value and present a genuine scientific contribution.

Another suggestion is to relate with strong justification and evidence the increase in the number of topics on the international level to a global move like climate change or the COVID pandemic outbreak …etc.

In section 3.2.1. keyword co-occurrence, please avoid describing the process of using the software, or the repeated definition of keyword co-occurrence to that already mentioned in table 1. ….Similarly section 3.2.2.

Please revise this figure caption.. Table 4. This is a table. Tables should be placed in the main text near to the first time they are cited.

Some of the references are not in English letters, please revise

I believe some sections in the results are very long, some of it can be summarized or moved to the literature review section.

minor revision required

Author Response

感谢您在百忙之中阅读这份手稿。
请参阅附件。

Reviewer 2 Report

I have thoroughly reviewed the manuscript, and I must say that it represents an outstanding contribution to the field. The paper is exceptionally well-structured, and all sections, from the introduction to the materials and methods, results, and discussion, are remarkably clear and well-written. The author(s) have done an exemplary job in presenting their research findings. I find no suggestions for improvement as the quality of the work is already exceptionally high.

This paper provides valuable insights into Therapeutic Landscapes and Community Open Spaces and will undoubtedly make a significant impact on future research in this area. I commend the author(s) for their meticulous research and the excellence in their presentation.

Best regards

Minor revisions needs to be done which can be solved by a proofread.

Author Response

Thank you very much for taking time out of your busy schedule to review this manuscript, and I'm glad that you fully appreciated this study. Your affirmation will be my motivation and confidence to continue to devote myself to research in the future.
Thank you again for your compliments.